# Cross-Sectional and Longitudinal Associations between Peak Expiratory Flow and Frailty in Older Adults

**DOI:** 10.3390/jcm8111901

**Published:** 2019-11-07

**Authors:** Caterina Trevisan, Debora Rizzuto, Stefania Maggi, Giuseppe Sergi, Anna-Karin Welmer, Davide Liborio Vetrano

**Affiliations:** 1Department of Medicine (DIMED), Geriatrics Division, University of Padova, 35128 Padova, Italy; giuseppe.sergi@unipd.it; 2Aging Research Center, Department of Neurobiology, Care Sciences and Society, Karolinska Institutet and Stockholm University, 17177 Stockholm, Sweden; debora.rizzuto@ki.se (D.R.); anna-karin.welmer@ki.se (A.-K.W.); davide.vetrano@ki.se (D.L.V.); 3Stockholm Gerontology Research Center, 113 46 Stockholm, Sweden; 4National Research Council, Neuroscience Institute, 35128 Padova, Italy; stefania.maggi@in.cnr.it; 5Division of Physiotherapy, Department of Neurobiology, Care Sciences and Society, Karolinska Institutet, 17177 Stockholm, Sweden; 6Allied Health Professionals, Function Area Occupational Therapy & Physiotherapy, Karolinska University Hospital, 14186 Stockholm, Sweden; 7Centro Medicina dell’Invecchiamento, IRCCS Fondazione Policlinico “A. Gemelli” and Catholic University of Rome, 00168 Rome, Italy

**Keywords:** peak expiratory flow, frailty, obstructive respiratory diseases, longitudinal study

## Abstract

Peak expiratory flow (PEF) has been linked to several health-related outcomes in older people, but its association with frailty is still unclear. This study investigates the association between PEF and prevalent and incident frailty in older adults. Data come from 2559 community-dwelling participants (age ≥ 60 years) of the Swedish National Study on Aging and Care in Kungsholmen (SNAC-K). Baseline PEF was expressed as standardized residual (SR) percentiles. Frailty was assessed at baseline and over six years, according to the Fried criteria. Associations between PEF and frailty were estimated cross-sectionally through logistic regressions, and longitudinally by multinomial logistic regression, considering death as alternative outcome. Obstructive respiratory diseases and smoking habits were treated as potential effect modifiers. Our cross-sectional results showed that the 10th–49th and <10th PEF SR percentile categories were associated with three- and five-fold higher likelihood of being frail than the 80th–100th category. Similar estimates were confirmed longitudinally, i.e., adjusted OR = 3.11 (95% CI: 1.61–6.01) for PEF SR percentiles < 10th, compared with 80th–100th percentiles. Associations were enounced in participants without physical deficits, and tended to be stronger among those with baseline obstructive respiratory diseases, and, longitudinally, also among former/current smokers. These findings suggest that PEF is a marker of general robustness in older adults, and its reduction exceeding that expected by age is associated with frailty development.

## 1. Introduction

The extension of life expectancy in middle- and high-income countries has led to the investigation of factors that may promote healthier aging. Frailty, a geriatric syndrome with increased vulnerability to stressors and reduced resilience, may hinder the achievement of this goal [1]. Therefore, special attention has been devoted to the early identification of individuals at higher risk of developing this syndrome, who could be targets of appropriate interventions. Toward this purpose, simple, easily accessible, and low-cost measures may be useful, especially when conducting studies on large cohorts or in clinical settings under time and resource constraints.

Peak expiratory flow (PEF) is a respiratory parameter that has been linked to negative health-related outcomes in older age, such as disability and mortality [2,3,4,5]. As a measure of respiratory function, PEF has lower diagnostic accuracy compared with spirometry, but it can be a valid alternative as first-line screening tool, and in older people who cannot undergo a thorough spirometry [6]. To a certain extent, a decline in PEF values is a normal physiological adaptation to aging. However, beyond a certain threshold, such decline may be associated with pathological conditions affecting cardiopulmonary fitness and expiratory function [6], as well as cognitive and physical performance [7,8]. As such, PEF can be considered as an index of general robustness that captures multiple aspects of individuals’ health across several organs and systems. Since frailty reflects the burden deriving from multiple biological impairments in an organism, a link between PEF and frailty can be hypothesized. Indeed, previous studies have found a bidirectional relationship between frailty and respiratory disorders characterized by both restrictive and obstructive patterns [9,10]. However, scanty evidence exists on the association between lung function as measured through PEF and frailty in older adults. Moreover, the few cross-sectional studies carried out so far are vitiated by the use of absolute PEF values [11,12], which may be a suboptimal parameter in older age [6,13].

We hypothesized that lower PEF values, exceeding the expected ones, might be associated with the presence and the development of frailty in the older population. The aims of our study were therefore to investigate the associations between PEF and the presence of frailty, and the risk of developing such syndrome in a large cohort of community-dwelling older adults.

## 2. Methods

### 2.1. Study Population

This study included data from the Swedish National Study on Aging and Care in Kungsholmen (SNAC-K), an ongoing prospective study that enrolled people aged ≥60 years. Study participants aged ≥78 years undergo follow-ups every third year, while the younger ones are assessed every six years. The participation rate at the baseline visit (2001 and 2004) was 73.3%, resulting in an initial sample of 3363 community-dwelling and institutionalized individuals.

For the present study, we excluded 191 institutionalized people, 339 who did not perform PEF measurement, 89 with missing anthropometric data, and 185 with missing information to define frailty at baseline, leaving a sample of 2559 older individuals for the cross-sectional analysis.

The longitudinal relationship between baseline PEF and incident frailty was investigated over a 6-year follow-up. For this analysis, 213 individuals who were frail at baseline, 55 who had incomplete information on frailty at follow-up, and 265 who were lost at follow-up were further excluded, leaving a sample of 2026 individuals. The comparison of characteristics between participants included and excluded due to missing data is reported in Appendix B.

SNAC-K was approved by the Regional Ethical Review Board in Stockholm (Sweden). Informed consent was obtained from all individual participants included in the study and, for participants with cognitive impairment, from the next of kin. The present study is reported following the STROBE guidelines.

### 2.2. Data Collection

Study participants underwent face-to-face interviews, clinical examinations, and testing by trained nurses and physicians to collect data on demographic characteristics, health behaviors, physical and cognitive performance, and medical conditions.

***PEF*** measurement was performed with a mini-Wright peak flow meter (Airmed Clement Clarke International^®^, Harlow, UK), which is a hand-held device for assessing the maximal speed of expiration of an individual. Participants were instructed to breathe in as deep as possible and then to blow as hard and fast as possible into the device, in a upright position. The highest value out of three tests was taken for the analyses and expressed as liters/minute [14]. In accordance with previous studies and in light of the physiological decline of PEF observed with aging and its inter-individual variability by age, sex, and body height, expected PEFs were estimated by normalizing PEF values by age, sex, and body height using a healthy subsample of the same population (i.e., people who had never smoked and had no diagnosis of respiratory disorders, cardiovascular diseases, or cancer) as reference group [13] (for details, please see Appendix A). PEF was operationalized as: (1) PEF residuals: defined as the difference between measured and expected PEF, considered as intervals decreasing by 10 L/min; (2) percent predicted: computed from the ratio between measured and expected PEF and considered as a discrete variable of decrement by 10%, and as a categorical one with cutoffs of 50%, 80%, and 100%; (3) PEF standardized residual (SR) percentiles: obtained from a normalization of the ratio (measured—predicted PEF)/(standard deviation of the residuals), where SR = 0 corresponds to the 50th percentile. SR percentiles were used as a continuous variable of decreasing 10th intervals, and categorized as <10th, 10th–49th, 50th–79th, and 80th–100th [13,15].

***Frailty*** was defined as the presence of at least three out of the following five criteria: weight loss, exhaustion, low physical activity, slow walking speed, and weakness [1]. Weight loss was defined as the loss of ≥1 kg in the last three months and exhaustion as the report of fatigue in the last three months. Low physical activity level was defined as the engagement in light or moderate/intense physical activity with a frequency of 2–3 times per month or less [16]. Light physical activities included walks on the sidewalk or paved surfaces, in parks, and/or in forests; short bike rides; light calisthenics; and golf. Moderate/intense physical activities were considered as brisk walking, jogging, heavy gardening, long bike rides, intense calisthenics, skating, skiing, swimming, ball games, or similar activities. Participants were classified as having a slow walking speed if they belonged to the lowest sex- and height-specific quintile of walking speed, measured over a distance of 6 m for those who considered themselves normal or fast walkers, or over a distance of 2.4 m for those who considered themselves slow walkers or for those who were assessed at home [17]. Participants lacking data on walking speed were considered as slow walkers if they reported to be unable to transfer at functional assessment (“Needs help in moving from bed to chair or requires a complete transfer”) [18]. Weakness was defined as the lowest quintile of handgrip strength, stratified by sex and body mass index (BMI) [1]. Handgrip was measured using a Grippit^®^ dynamometer (Catell AB, Hägersten, Sweden), following standard procedures. Handgrip was tested in both hands, and the best of the two tests was used in the analyses. Participants lacking data on handgrip strength were considered to be weak if they reported to be unable to perform a simple manual activity (i.e., open a jar that has a screw-on lid).

***Covariates.*** Participants’ educational level was classified on the basis of the highest level of formal education as elementary, high school, and university or above. Body Mass Index (BMI) was computed as the ratio of participants’ body weight (kg) by height squared (m^2^). We categorized smoking habits as never, former, and current smoking; and alcohol consumption as no or occasional, light to moderate (1–14 drinks/week for men and 1–7 drinks/week for women), or heavy (≥15 drinks/week for men and ≥8 drinks/week for women) consumption. Cognitive performance was assessed by physicians using the Swedish version of the Mini Mental State Examination (MMSE), and the presence of cognitive deficits was defined as a MMSE score <28 [19]. Physical function was examined by balance (standing on one leg), chair stand, and walking speed tests. The presence of physical deficits was defined as the inability to perform at least one of the following: standing on one leg for ≥5 s, performing 5 consecutive chair stands without using the arms, walking at a speed ≥0.8 m/s [20]. On the basis of participants’ clinical examination, medication use, biochemical analyses, and medical history, physicians evaluated the presence of obstructive respiratory diseases, namely chronic obstructive pulmonary disease (COPD) or asthma; and cardiovascular diseases (CVD), defined as at least one among ischemic heart disease, heart failure, and atrial fibrillation. The total number of chronic diseases (excluding CVD, COPD, and asthma) was also assessed, considering a disease as chronic if it was of prolonged duration and (1) left residual disability or worse quality of life; or (2) required a long period of care, treatment, or rehabilitation [21]. Moreover, information on inhaled bronchodilators use (ATC codes: R03BA, R03AK, R03AL08, R03AL09) was recorded.

Date and cause of death were obtained from the Swedish Cause of Death Registry.

### 2.3. Data Analyses

Characteristics were compared across individuals by PEF SR percentile intervals through ANOVA and the Chi-squared test, as appropriate. The frequency of missing covariate values was <5%, so we performed a single imputation using the expectation maximization algorithm for continuous variables, while dummy variables were used for the categorical ones. The cross-sectional analysis on the association between PEF and frailty at baseline was evaluated through logistic regression. The longitudinal relationship between PEF and incident frailty over the 6-year follow-up in participants non-frail at baseline was evaluated through multinomial logistic regression. In the latter analysis, participants who reached the follow-up assessment and did not become frail represented the reference category, while incident frailty at follow-up or death before the follow-up assessment were considered alternative outcomes. Both cross-sectional and longitudinal analyses were first adjusted by socio-demographic characteristics, and second also for those variables differently distributed between PEF SR percentile groups and that may act as potential confounders. In the longitudinal analysis, we also adjusted for the presence of physical deficits at baseline, and for study participation time, considering time to frailty (for incident frailty cases), time to the last available follow-up (for those non-frail at the follow-up assessment), or time to death (for those who died before the follow-up assessment). The strength of these associations was expressed as odds ratios (ORs) with 95% confidence intervals (95% CIs). As a sensitivity analysis, we evaluated the association between PEF and incident frailty only in participants with no physical deficits at baseline, in order to select the healthiest individuals. To investigate whether the association between PEF and frailty was modified by the presence of obstructive respiratory diseases (COPD or asthma) at baseline, and by smoking habits (never vs. former/current), we tested interactions by including the multiplicative interaction term in the fully adjusted model, and we performed appropriate subgroup analyses. All statistical tests were two-tailed, and statistical significance was set as a *p*-value < 0.05. Analyses were performed using IBM SPSS Statistics for Windows, Version 23.0. (Armonk, NY, United States).

## 3. Results

The baseline characteristics of the 2559 participants are shown in Table 1. The mean age (±SD) of the study participants was 71.7 ± 9.7 years, and 61.3% were women. More than half of the sample reported former (39.7%) or current (15%) smoking habits, 4.1% had a diagnosis of COPD, and 6.3% of asthma. Compared with individuals included in the highest PEF SR percentile groups, those with lower values were more likely to have lower educational level, lower BMI, and to be current smokers. Concerning health status, the lower the PEF SR percentile, the higher the number of chronic diseases, the prevalence of cardiovascular and respiratory diseases, bronchodilator use, and deficits in cognitive or physical functions. At baseline, 213 (8.3%) participants were frail. The prevalence of frailty significantly increased from the highest to the lowest PEF SR percentile group (4.2%, 5.2%, 9.1%, and 19.4%, respectively; *p* < 0.001).

In cross-sectional analyses, the odds of presenting frailty increased by around 20% per each 10-unit decrease in PEF expressed either as SR percentile (OR = 1.23, 95% CI: 1.15–1.30) or as percent predicted (OR = 1.21, 95% CI: 1.14–1.29), and were more than three- and five-fold higher for the participants in the lowest PEF categories (10th–49th and <10th SR percentiles, respectively), compared with the highest ones (Table 2, Model 2).

Over 6 years, 276 participants (10.8%) died and 227 (8.9%) developed frailty. The odds of incident frailty increased by around 16% per each 10-unit decrease in PEF SR percentile. Moreover, compared with the highest PEF category, the odds ratio of incident frailty was more than twice as high for those with PEF SR percentiles in the range 10th–49th, and more than three-fold higher for the <10th PEF SR percentile group. Similar results were observed when PEF was expressed as percent predicted (Table 3, Model 2). The association between PEF and incident frailty was even stronger in the sensitivity analysis where only individuals free from physical deficits at baseline were included (Appendix A).

A marginally significant interaction was observed between baseline PEF SR percentiles groups and obstructive respiratory diseases (diagnosis of COPD and/or asthma) in influencing the risk of developing frailty at follow-up (*p_interaction_* = 0.05). After stratifying analyses by obstructive respiratory disease (Table 4), the association between PEF and frailty tended to be stronger in people with obstructive respiratory disease compared with their counterparts.

Finally, when exploring the same relationships in an analysis stratified by smoking habit, we observed no significant differences between groups in the cross-sectional analysis, while a statistically significant association of baseline PEF with incident frailty emerged only among those who reported former or current smoking habit (*p_interaction_* = 0.003; Table 5).

## 4. Discussion

Our study suggests that PEF values lower than the expected were associated with both prevalent and incident frailty in older adults. The presence of COPD and asthma, as well as of former or current smoking habit, may magnify this association.

The relationship between frailty and respiratory function as estimated through PEF has been explored by previous studies, which reported lower PEF values in pre-frail and frail older individuals, compared with non-frail ones [9,11,12]. These findings suggest that PEF may be considered as an indicator of individuals’ robustness, and the inclusion of this parameter in various proposed frailty indices further supports this view [3,8,22]. However, the cross-sectional nature [11,12] or the setting-specific results (e.g., nursing home) [23] of previous works limit the evaluation of the role of PEF in the identification of the most vulnerable individuals for the development of frailty in the general older population, as suggested by our study.

The association between PEF and frailty can be supported by several mechanisms. Among these, sarcopenia—that is, the loss of muscle strength, quantity, and quality that characterizes the aging process [24]—may involve both appendicular and respiratory skeletal muscle [25]. Sarcopenia is a key player in most of the criteria proposed by Fried and colleagues for the operationalization of physical frailty (e.g., weakness and slowness) [1,26,27]. At the same time, acting on inspiratory and expiratory muscles, sarcopenia may eventually accelerate the age-related changes in the respiratory dynamics [6]. These include the reduction in the chest wall compliance and the loss of respiratory muscle mass and function [6], which can result in a decrease by up to a quarter of diaphragm strength [28]. By these processes, sarcopenia can partially explain a reduction in PEF, which is actually an effort-dependent measure [29]. Ultimately, such respiratory dysfunction may be a further determinant of fatigue, low physical performance and decreased physical activity level [30,31,32,33], promoting functional decline [32] and falls [34], and increasing the overall risk of frailty in a vicious circle. Among others, a mechanism that could underpin both respiratory impairment and frailty is chronic inflammation [1,35]. In particular, given that both frailty and respiratory dysfunction may be manifestations of the aging process, the role played by a pro-inflammatory environment (i.e., inflammaging) in the development of such conditions may be hypothesized. In fact, inflammaging may contribute to accelerate muscle waste on one hand, and parenchymal lung damage on the other [27,36,37]. However, it would be difficult to untangle the role played by inflammaging and that of concomitant chronic diseases in the exacerbation of frailty and poor lung function. In addition, the impairment of expiratory and inspiratory functions can result in a dysregulation of cough reflex, favoring the onset of respiratory infections [38]. Low PEF could thus indirectly embody the greater vulnerability of individuals to respiratory infections, which have a strong effect on global health status [39] and on the development of frailty. Finally, poor cardiopulmonary performance may affect cognitive performance, and low PEF has been previously associated with worse cognitive functions and higher dementia risk [7,40,41,42]. The pathophysiological pathways under this association are still unclear, but chronic hypercapnia or hypoxemia conditions may play a role in this regard, with an impact on cognitive functions [43] and on frailty onset.

Although the abovementioned mechanisms support a bidirectional link between respiratory function and frailty [9], our longitudinal analyses suggest that low PEF may be a determinant, as well as an early predictor, of frailty. In this regard, the stronger results observed in participants free from physical deficits at baseline strengthen the potential usefulness of PEF in the detection of individuals most vulnerable to frailty. The association between PEF and frailty appeared more marked among people with obstructive respiratory disease, displaying almost doubled risk estimates both for the presence of frailty at baseline and for its development over the follow-up, in line with previous works [10]. However, after stratifying the sample by smoking habits the probability of presenting frailty as a function of PEF was similar in older adults who never or ever smoked, while the association between PEF and frailty development was stronger among former or current smokers. These results suggest that, despite equal PEF performance, the presence of respiratory symptoms—which are likely to characterize individuals with COPD or asthma diagnosis—as well as the exposure to the toxic effects of smoke could increase the burden of respiratory dysfunction and make individuals more vulnerable to developing frailty. We acknowledge that this issue needs to be explored in further investigations with larger samples of patients with obstructive respiratory diseases, taking into consideration the respiratory function and the possible benefit linked to the use of specific drugs. However, such data support the potential importance of an early identification and treatment of conditions with respiratory impairment and related symptoms in order to attenuate their impact on frailty development.

Among the limitations of this work, the unavailability of formal spirometry data and the possible under-diagnosis of clinical conditions with airway obstruction may have been sources of misclassification bias, limiting the reliability of our subgroup analyses. However, the ascertainment of COPD and asthma in our study was performed based on multisource information from medical records and physical examination of study participants [21], thus we can confidently argue that individuals with low PEF and no diagnosis of lung disorders may have had respiratory impairment with no clear disease-specific symptoms. Moreover, the individuals excluded from our analytical sample due to missing information or dropouts were more likely to be older and to have worse health status, leading to a possible underestimation of both prevalent and incident cases of frailty. On the other hand, the large sample size of community-dwelling older adults with reliable data on PEF and frailty, the use of different approaches to evaluate PEF, and the longitudinal study design represent strengths of our work.

In conclusion, this study suggests that normal PEF values may be considered as an indicator of general robustness, and that PEF values lower than those expected by age are associated with frailty development. Future investigations are needed to evaluate whether the use of pharmacological and non-pharmacological interventions of chronic respiratory diseases may attenuate the association between respiratory function and frailty.

## Figures and Tables

**Table 1 jcm-08-01901-t001:** Overall participant baseline characteristics and by peak expiratory flow (PEF) standardized residual percentiles.

Characteristics	All (*n* = 2559)	Peak Expiratory Flow SR Percentiles			
		80th–100th (*n* = 333)	50th–79th (*n* = 892)	10th–49th (*n* = 1025)	<10th (*n* = 309)
Age (years)	71.7 ± 9.7	74.4 ± 9.8	71.1 ± 9.3	70.9 ± 9.6	73.4 ± 10.2 ***
Sex (female)	1568 (61.3)	221 (66.4)	554 (62.1)	605 (59.0)	188 (60.8)
Educational level					
Elementary	356 (13.9)	43 (12.9)	99 (11.1)	146 (14.2)	68 (22.0) ***
High school	1229 (48.0)	173 (52.0)	423 (47.4)	488 (47.6)	145 (46.9)
University	973 (38.0)	117 (35.1)	370 (41.5)	391 (38.1)	95 (30.7) **
Body mass index (kg/m^2^)	25.8 ± 4.0	26.2 ± 3.9	26.0 ± 3.7	25.8 ± 4.2	25.2 ± 4.4 **
Alcohol consumption					
No or occasional	751 (29.3)	108 (32.4)	223 (25.0)	292 (28.5)	128 (41.4) ***
Light to moderate	1533 (59.9)	193 (58.0)	574 (64.3)	619 (60.4)	147 (47.6) ***
Heavy	269 (10.5)	32 (9.6)	93 (10.4)	110 (10.7)	34 (11.0)
Smoking habits					
Never	1144 (44.7)	169 (50.8)	429 (48.1)	433 (42.2)	113 (36.6) ***
Former	1016 (39.7)	134 (40.2)	355 (39.8)	419 (40.9)	108 (35.0)
Current	385 (15.0)	29 (8.7)	106 (11.9)	163 (15.9)	87 (28.2) ***
Number of chronic diseases	3.6 ± 2.2	3.6 ± 2.0	3.4 ± 2.1	3.6 ± 2.2	4.1 ± 2.4 ***
COPD	105 (4.1)	5 (1.5)	18 (2.0)	38 (3.7)	44 (14.2) ***
Asthma	160 (6.3)	11 (3.3)	47 (5.3)	71 (6.9)	31 (10.0) **
CVD	485 (19.0)	56 (16.8)	135 (15.1)	214 (20.9)	80 (25.9) ***
Use of bronchodilators	145 (5.7)	8 (2.4)	49 (5.5)	58 (5.7)	30 (9.7) **
Cognitive deficits	320 (12.5)	35 (10.5)	88 (9.9)	122 (11.9)	75 (24.3) ***
Walking speed (m/s)	1.10 ± 0.36	1.15 ± 0.33	1.16 ± 0.34	1.10 ± 0.37	0.92 ± 0.38 ***
Physical deficits	877 (34.3)	108 (32.4)	254 (28.5)	340 (33.2)	175 (56.6) ***

Values are presented as mean ± standard deviation or absolute number and percentage (%). *Abbreviations:* SR, standardized residual; COPD, chronic obstructive pulmonary disease; CVD, cardiovascular diseases. *Notes*: cognitive deficits were defined as a Mini-Mental State Examination score <28; physical deficits were defined as the inability to perform at least one among standing on one leg for ≥5 s, performing 5 consecutive chair stands without using the arms, or walking at a speed ≥0.8 m/s. ** *p*-value < 0.01; *** *p*-value < 0.001.

**Table 2 jcm-08-01901-t002:** Cross-sectional association between peak expiratory flow and frailty.

PEF Measures	*n*	Odds Ratios and 95% Confidence Intervals of Frailty	
		Model 1	Model 2
**PEF Residuals**			
Per each 10 L/min decrease	2559	1.07 (1.05–1.08) ***	1.06 (1.04–1.07) ***
**PEF SR Percentile**			
Per each 10th decrease	2559	1.26 (1.19-2.34) ***	1.23 (1.15–1.30) ***
80th–100th	333	1.00 (ref)	1.00 (ref)
50th–79th	892	1.84 (0.97–3.48)	1.79 (0.93–3.44)
10th–49th	1025	3.55 (1.94–6.51) ***	3.03 (1.63–5.66) ***
<10th	309	7.49 (3.94–14.24) ***	5.79 (2.95–11.35) ***
**PEF Percent Predicted**			
Per each 10% decrease	2559	1.26 (1.19–2.33) ***	1.21 (1.14–1.29) ***
>100%	1225	1.00 (ref)	1.00 (ref)
80%–100%	870	1.71 (1.15–2.53) **	1.50 (0.99–2.25)
50%–79%	376	4.44 (2.99–6.61) ***	3.81 (2.50–5.81) ***
<50%	88	6.10 (3.48–10.69) ***	4.43 (2.43–8.08) ***

Model 1 is adjusted for age and sex. Model 2 is also adjusted for educational level, body mass index, smoking habits, drinking habits, Mini-Mental State Examination score, number of chronic diseases, chronic obstructive pulmonary disease, asthma, cardiovascular diseases, and use of bronchodilators. *Abbreviations:* PEF, peak expiratory flow; SR, standardized residual. * *p*-value < 0.05; ** *p*-value < 0.01; *** *p*-value < 0.001.

**Table 3 jcm-08-01901-t003:** Longitudinal association between peak expiratory flow and frailty over a 6-year follow-up.

PEF Measures	*n*	Odds Ratios and 95% Confidence Intervals of Frailty	
		Model 1	Model 2
***PEF residuals***			
Per each 10 L/min decrease	2026	1.06 (1.04–1.07) ***	1.05 (1.03–1.07) ***
***PEF SR percentile***			
Per each 10th decrease	2026	1.19 (1.12–2.26) ***	1.16 (1.09–1.23) ***
80th–100th	283	1.00 (ref)	1.00 (ref)
50th–79th	750	1.59 (0.95–2.64)	1.43 (0.84–2.41)
10th–49th	783	2.63 (1.59–4.35) ***	2.27 (1.35–3.83) **
<10th	210	4.15 (2.24–7.71) ***	3.11 (1.61–6.01) **
***PEF percent predicted***			
Per each 10% decrease	2026	1.24 (1.15–2.33) ***	1.20 (1.11–1.29) ***
>100%	1033	1.00 (ref) [ref]	1.00 (ref)
80%–100%	686	1.65 (1.15–2.35) **	1.55 (1.07–2.25) *
50%–79%	259	2.99 (1.93–4.64) ***	2.56 (1.59–4.10) ***
<50%	48	4.24 (1.80–10.03) **	3.35 (1.35–8.30) **

Model 1 is adjusted for age, sex, and study time (time to frailty/to follow-up/to death, as appropriate). Model 2 is also adjusted for educational level, body mass index, smoking habits, drinking habits, baseline Mini-Mental State Examination score, physical deficits at baseline, number of chronic diseases, chronic obstructive pulmonary disease, asthma, cardiovascular diseases, and use of bronchodilators. *Abbreviations:* PEF, peak expiratory flow; SR, standardized residual. * *p*-value < 0.05; ** *p*-value < 0.01; *** *p*-value < 0.001.

**Table 4 jcm-08-01901-t004:** Association between peak expiratory flow and frailty in participants stratified by the presence of obstructive respiratory diseases (chronic obstructive pulmonary disease and/or asthma).

PEF Measures	Odds Ratios (95% Confidence Intervals) of Frailty	
	No Obstructive Respiratory Disease	Obstructive Respiratory Disease
***Cross-sectional analysis***		
***n***	2318	241
**PEF residuals**		
Per each 10 L/min decrease	1.06 (1.04–1.08) ***	1.08 (1.02–1.13) **
**PEF SR percentile**		
Per each 10th decrease	1.21 (1.13–1.29) ***	1.46 (1.16–1.84) **
**PEF percent predicted**		
Per each 10% decrease	1.21 (1.13–1.29) ***	1.35 (1.10–1.66) **
***Longitudinal analysis***		
***n***	1843	183
**PEF residuals**		
Per each 10 L/min decrease	1.04 (1.02–1.06) ***	1.10 (1.03–1.17) **
**PEF SR percentile**		
Per each 10th decrease	1.14 (1.07–1.22) ***	1.38 (1.09–1.74) **
**PEF percent predicted**		
Per each 10% decrease	1.17 (1.08–1.27) ***	1.49 (1.15–1.94) **

Model adjusted for age, sex, educational level, body mass index, smoking habits, drinking habits, baseline Mini-Mental State Examination score, number of chronic diseases, cardiovascular diseases, and use of bronchodilators. Longitudinal analyses are also adjusted for study time and for physical deficits at baseline. *Abbreviations:* PEF, peak expiratory flow; SR, standardized residual. ** *p*-value < 0.01; *** *p*-value < 0.001.

**Table 5 jcm-08-01901-t005:** Association between peak expiratory flow and frailty in participants stratified by smoking habits.

PEF Measures	Odds Ratios (95% Confidence Intervals) of Frailty	
	Never Smokers	Former or Current Smokers
***Cross-sectional analysis***		
***n***	1144	1401
**PEF residuals**		
Per each 10 L/min decrease	1.06 (1.03–1.09) ***	1.05 (1.03–1.08) ***
**PEF SR percentile**		
Per each 10th decrease	1.23 (1.13–1.34) ***	1.22 (1.11–1.34) ***
**PEF percent predicted**		
Per each 10% decrease	1.21 (1.11–1.32) ***	1.21 (1.11–1.33) ***
***Longitudinal analysis***		
***n***	907	1107
**PEF residuals**		
Per each 10 L/min decrease	1.02 (0.99–1.05)	1.07 (1.04–1.09) ***
**PEF SR percentile**		
Per each 10th decrease	1.05 (0.96–1.15)	1.28 (1.17–1.40) ***
**PEF percent predicted**		
Per each 10% decrease	1.08 (0.97–1.20)	1.34 (1.20–2.50) ***

Model adjusted for age, sex, educational level, body mass index, smoking habits (current vs. former, only for the subgroup analysis on ever smokers), drinking habits, baseline Mini-Mental State Examination score, number of chronic diseases, chronic obstructive pulmonary disease, asthma, and cardiovascular diseases. Longitudinal analyses are also adjusted for study time and for physical deficits at baseline. *Abbreviations:* PEF, peak expiratory flow; SR, standardized residual. *Notes*: 14 individuals were excluded from the analyses because of missing data on smoking habits. *** *p*-value < 0.001.

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
