# Peer review of "Cross-Sectional and Longitudinal Associations between Peak Expiratory Flow and Frailty in Older Adults"

_jcm, 2019, doi:10.3390/jcm8111901_

Round 1

Reviewer 1 Report

This is a well written paper which describes the relationship between peak flow rate and patient frailty. I largely agree with the authors’ presentation, however some discussion of the study is needed.

General comments

In patients without airway obstruction or COPD, PEF is reflective of respiratory muscle strength. I think that the authors should investigate correlations among declining PEF, frailty state, respiratory function, respiratory muscle strength, and aging. I think respiratory muscle strength is particularly important.

In previous studies, peak cough flow (related with PEF) has been given as an absolute value, because the flow rate required to clear a foreign substance from the airway is determined by absolute flow rate. It might be better for the authors of this paper to investigate the relationship of between absolute PEF value and frailty.

Specific comments

The ratio of subjects with FEV1.0<70 needs to be included in Table 1 In model 1 of the statistical analysis of each group of subjects, educational level, age, and sex were used as covariates. However, I think that minimum covariates should be used in this model. In Line 267, the authors refer to the influence of chronic inflammation on PEF values. But I am concerned whether it is explained correctly. As per my understanding, the reduction of PEF values is mainly caused by airway obstruction, and the latter is mainly caused by smoking. Both sarcopenia and COPD correlate with chronic inflammation, but those mechanisms might differ from that suggested here.

Reviewer 2 Report

In „Cross-sectional and longitudinal associations between peak expiratory flow and frailty in older adults” Trevisan et al. assess the relation between peak expiratory flow and prevalent as well as incident frailty in the general population. The analyzed data from 2559 older adults from the SNAC-K study. The authors used logistical and multinomial logistic regression models for the prevalent and incident analyses, respectively. They report that low PEF is associated with a higher risk of frailty in the cross-sectional and longitudinal setting. While these data are interesting, I have some concerns regarding the methodology used.

- A major concern is that the authors created their own reference values based on their own definition of “health” (i.g. no diagnosis of respiratory diseases, cardiovascular disorders or cancer). Without showing the progression of the ageing related decline in PEF. This information is essential for the reader to understand the quantifiable difference between non-frail and frail individuals of similar height, sex and age.

- A second major concern is that throughout the manuscript the authors imply that subjects with low PEF may help to identify frailty. Even if this may be true, the authors did not test. Further, in the introduction the authors even state that “of our study were therefore to investigate 62 the associations between PEF and the presence of frailty, and the risk of developing such syndrome 63 in a large cohort of community-dwelling older adults.” In addition, the last sentence of the abstract concludes that “These findings suggest that PEF is a marker of general robustness in older adults and its reduction, exceeding that expected by age, predicts frailty development.” This needs to be corrected so that their conclusions are based purely on the findings of this analysis. Since the authors already defined frailty based on five criteria (weight loss, exhaustion, low physical activity, slow walking speed, and weakness) the question becomes whether PEF adds any additional information – AIC would an appropriate tool for this analysis.

There are several minor issues with the submission:

- The introduction is poorly organized and does, for example, not contain any information regarding the age and sex distribution of the study population.

- It is unclear whether all group comparisons were tested or if the planned comparisons were planned tests. In either case adjustment for multiple testing is most likely necessary. Albeit the opinion of a statistician would be helpful in this matter.

- In line 73 you write “613 with missing data on PEF assessment (n = 339).” Please explain.

- Please provide more information for the device used to measure PEF (city and country is missing)

Round 2

Reviewer 2 Report

The authors have adressed all of my concerns. Thank you!